# Nanostructuration Impact on the Basic Properties of the Materials: Novel Composite Carbon Nanotubes on a Copper Surface

Natalia Kamanina [1,2,3,4,*], Andrei Toikka [3,4] and Dmitry Kvashnin [5,6,7]

1 Lab for Photophysics of Media with Nanoobjects, Vavilov State Optical Institute, Kadetskaya Liniya V.O., dom 5/2, 199053 St. Petersburg, Russia
2 Lab for Photophysics of Nanostructured Materials and Devices, Vavilov State Optical Institute, Kadetskaya Liniya V.O., dom 5/2, Babushkina str., dom 36, korp.1, 192171 St. Petersburg, Russia
3 Department of Photonics, St.-Petersburg Electrotechnical University ("LETI"), Ul. Prof. Popova, dom 5, 197376 St. Petersburg, Russia; atoikka@obraz.pro
4 Department of Advanced Development, Petersburg Nuclear Physics Institute, National Research Center "Kurchatov Institute", 1 md. Orlova Roshcha, 188300 Gatchina, Russia
5 Emanuel Institute of Biochemical Physics RAS, 4 Kosigina st., 119334 Moscow, Russia; dgkvashnin@phystech.edu
6 Department of Physics and Mathematics, Pirogov Russian National Research Medical University, Ostrovitianov Str. 1, 117997 Moscow, Russia
7 School of Chemistry and Technology of Polymer Materials, Plekhanov Russian University of Economics, Stremyanny Lane, 36, 117997 Moscow, Russia
* Correspondence: nvkamanina@mail.ru; Tel.: +7-(812)-327-00-95

**Abstract:** Copper is important material that is widely applicable in the electric and electronic industries. Nevertheless, in some circumstances, it is highly desirable to improve its properties. Therefore, combination of materials of various composition and properties attracts scientific and industrial society. Here, the composite based on carbon nanotubes (CNTs) on a Cu surface was fabricated using laser-oriented deposition (LOD) technique and studied. Examination of the novel composite showed that its reflectance was decreased, the microhardness was increased, and wetting of the surface exhibited higher hydrophobicity. A molecular dynamic simulation showed that the penetration depth increases with nanotube diameter decrease and growth of the acceleration rate. Topography observations made via AFM images revealed a dense thin film with an almost-homogeneous distribution of CNTs, with several locations with irregular thickness addressing the different lengths of CNTs.

**Keywords:** nanostructuration process; composite properties change; copper; carbon nanotubes; laser-oriented deposition technique; molecular dynamics

## 1. Introduction

The nanostructuration process plays an important role in the change of the basic physical-chemical properties of the materials. Not only can the volume of the matrix material be structured, but also its surface. Among the different nanoparticles and nanofibers, carbon nanotubes (CNTs) are more often use for this aim. CNTs have been attracting scientific and industrial interest around the world since their discovery at the end of 20th century. CNTs exhibit extensive mechanical, chemical and physical properties. They exhibit high thermal conductivity [1–3] and unique mechanisms of the charge carrier motion [4]. CNTs also provide an excellent strength performance due to the strong C-C bonds [5,6]; the Young's modulus of the C-C bond is close to the experimentally determined value of 0.32–1.47 TPa, while the theoretically predicted values vary at the range of 0.5–5.5 TPa. Additionally, they have low refractivity [7–10] with the refractive index close to 1.05–1.1 that makes them transparent when applied on the materials' surface.

Due to their extensive performance, the CNTs have a high scientific and industrial interest in combination with other structural materials, including copper. Copper is one of the engineering materials that demonstrate a high electric and thermal conductivity, high corrosion resistance, and high mechanical performance. Copper and copper alloys are usually used in the electric and electronic devices [11–18]. Indeed, the high conductivity of copper makes it an extremely interesting object of the application for electrical engineering and optoelectronics. Therefore, different types of the treatments that modify the basic parameters of copper are interesting and relevant.

Hence, the production of the structural material that combines the copper matrix and CNTs may ensure its advanced performance. Therefore, the fabrication of copper with CNTs films became an attractive approach. Crespi et al. reported carbon deposition on copper substrate using magnetron deposition technique [19]. Acauan et al. described method of direct synthesis of nanomaterials on bulk copper using chemical vapor deposition (CVD) [20]. It should be mentioned that other materials have been treated with the CNTs with good advantage. Van-Duong Dao et al. showed the development of the new hybrid systems based on MWNT-Pt for the use of dye-sensitized solar cells [21]; moreover, this team has extended the variation of the Pt-based materials via construction of the Pt-based alloy/graphene nanohybrid for photovoltaics [22]. Kamanina et al. extensively investigated laser-oriented deposition (LOD) technology for production of structural materials with the advanced surface performance [23–27]. KBr [23], LiF [24], Al [25] have been studied and their physical-chemical properties have been improved via application of the laser technique. In each of these cases of the material processing, the ratio between the parameters of the elementary lattice of the matrix and the diameter of the nanotubes was taken into account; the expansion rate of the carbon nanotubes was also taken into account as well when an electric field was applied in the range of 100–600 V/cm. It should be noticed that the LOD is a versatile method of the thin film deposition with several advantages over other methods as high deposition rate, production of highly uniform morphology, and obtaining coating that has no contamination.

In this paper, a fabrication of a thin film of CNT on a copper substrate using the LOD method is shown. An examination of the spectra, surface wettability and microhardness were carried out on the deposited copper and compared with an original one. The molecular dynamic simulation of the process was conducted to evaluate behavior of the CNT deposition process on copper substrate. Topography of the obtained surface was characterized by atomic force microscopy (AFM).

## 2. Materials and Methods

The production of the CNT thin film on the copper substrate by the LOD method was carried out using the IR $CO_2$-laser operated at the wavelength of 10.6 µm with the power of 30 W and a beam diameter of 5 mm, which was connected to a vacuum chamber. The detailed description of the LOD process can be found in [23] and the patents in [26,27]. In order to briefly explain our procedure, the variant of the scheme used in the current study is shown in Figure 1. Orienting the grid permitted the placement of the CNTs in the vertical position.

It should be stated that the vertical orientation of CNTs was achieved by an electric field of 100–600 V·cm$^{-1}$. We used the single-walled CNTs (SWCNTs) with the average diameter of 0.7–0.83 nm and a specific surface area $\geq$700 m$^2$/g (Sigma-Aldrich Co., Saint Louis, MO, USA) and Cu substrate with a thickness of 0.5 mm and diameter of 20 mm.

The reflectance spectra were collected using the Perkin-Elmer Lambda 9 (PerkinElmer Optoelectronics, Waltham, MA, USA) and Furrier FSM-1202 device (Nica-Garant+, Saint-Petersburg, Russia). Microhardness Vickers tests were carried out by PMT-3M tester (LOMO Ltd., Saint-Petersburg, Russia). The microhardness values illustrated in the work are the average of five points on each sample. The contact angle (CA) was determined using the OCA 15EC device (LabTech Co., Saint-Petersburg-Moscow, Russia). Contact angle values presented in the work are the average of five points on each sample. AFM

topography images were carried out in an area of 50·50 μm² using Solver Next AFM (NT-MDT LLC, Moscow, Russia).

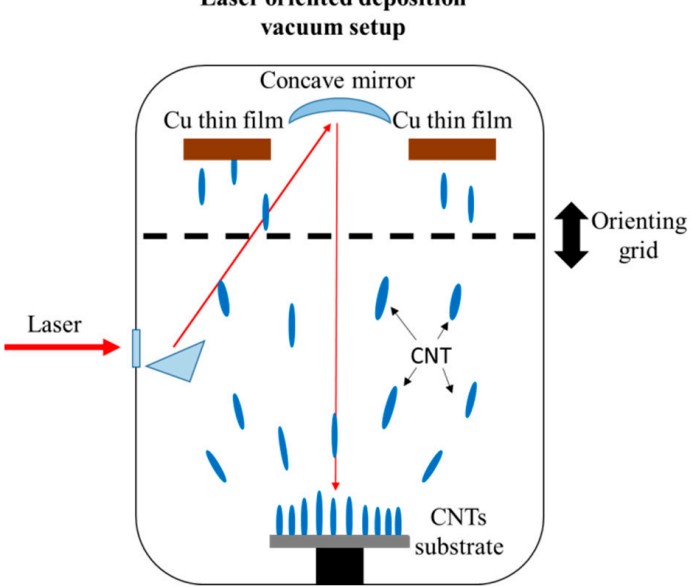

**Figure 1.** The laser-oriented deposition scheme used for the Cu-surface modification.

Modeling of the LOD process was performed by the molecular-dynamics simulation using LAMMPS package [28]. The total time of simulation was 60 ps with the time step of 1 fs and the acceleration rate varied from 100 to 600 m/s. Interactions between the Cu atoms and CNTs were described with embedded atom method (EAM) [29]. Behavior of the CNT at finite temperature was illustrated by the theoretical approach of adaptive intermolecular reactive empirical bond order potential (AIREBO) [30]. Interactions between CNT and Cu substrate was defined by 12–6 potential in the Lennard–Jones form and a behavior of the system at a constant temperature of 300 K was performed by means of Nosé–Hoover thermostat [31,32].

## 3. Results and Discussion

The reflectance spectra of the CNTs-deposited surface and original copper are shown in Figure 2. The deposition was carried out by specular reflection at 45°.

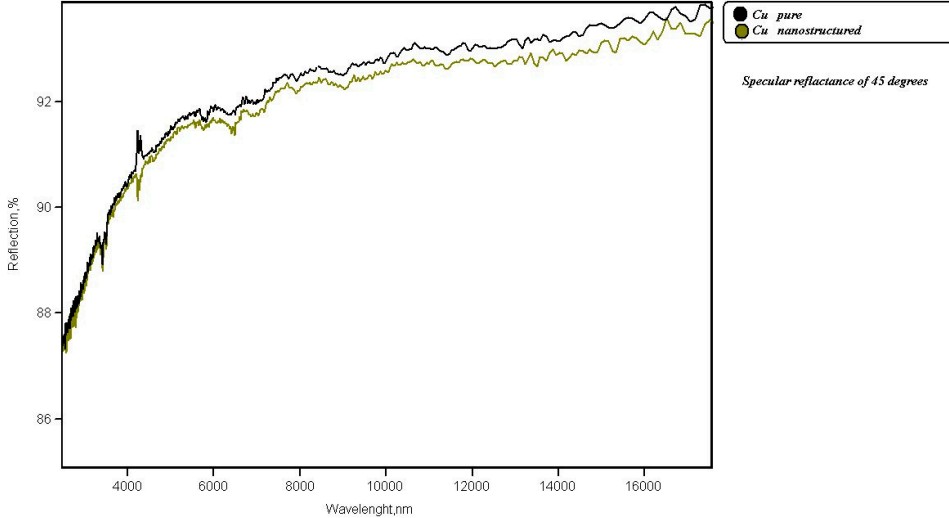

**Figure 2.** Reflectance spectra of the CNTs-deposited surface and the original copper.

The change in the reflectance spectra can be explained via the refractive features of the CNTs deposited on the Cu surface. The refractive index of copper, at the range of visible light, is $n = 1.06$ at the wavelength of 500 nm, and at the range of near infrared, is $n = 3.1$ at the wavelength of 5000 nm [33]. The low refractive index of the CNTs predicts the decrease of the Fresnel losses of the novel composite. Figure 2 shows two curves of the reflectance for an original copper and a sample with a thin film of CNTs. The spectra collected at the wavelength range of 3000–17,000 nm. Evaluation of the obtained spectral data revealed that the reflectance decreases partially with the wavelength growth.

Microhardness test results of the examined samples are listed in Table 1.

**Table 1.** Microhardness of the original and modified by copper CNTs.

| Material | Microhardness Average [MPa] | Increase [%] |
|---|---|---|
| Original copper | 1026 | |
| Copper + CNTs | 1118 | ~9 |

As expected, microhardness measurements revealed the increase of the values for the modified sample. The average microhardness of the original copper was 0.1026 GPa while it reached 0.1118 GPa for the modified sample. The CNTs exhibited high mechanical properties, therefore the increase in microhardness of copper may be addressed to their presence on the samples' surface. These data are in good accordance with the wetting angle estimation.

An examination of the wettability of the obtained thin film was carried out using the contact angle (CA) measurements. The obtained images are shown in Figure 3.

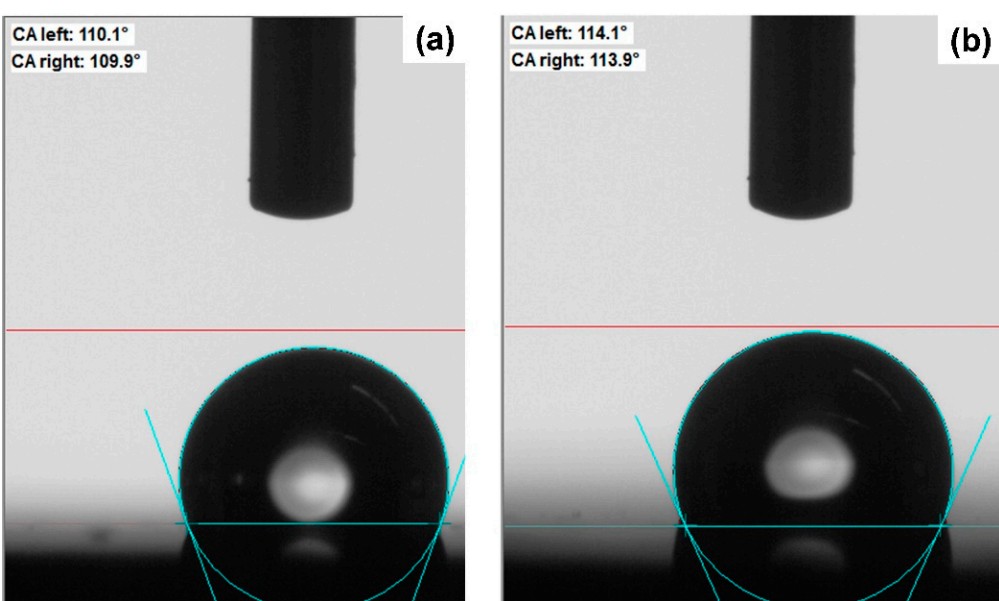

**Figure 3.** The CA of (**a**) original copper and (**b**) CNTs-deposited surface.

As observed in Figure 3, the CA of the original alloy exhibits an average value of 110° and the surface with the deposited CNTs demonstrates an average value of 114°. The higher hydrophobicity of the obtained thin film is addressed to the Lotus effect provided by the appearance of the CNTs on the surface. These results are well-correlated with the results shown before on the CNTs-deposited surface into the KBr, LiF, and Al materials [23–25].

The molecular dynamic simulation process was conducted to characterize the behavior of the CNTs interaction with copper substrate. The simulation was done using two types of CNTs and two acceleration rates. The simulation shown in Figure 4 introduces CNTs with diameter of 0.94 and 2.68 nm and acceleration rates of 100 and 600 m/s.

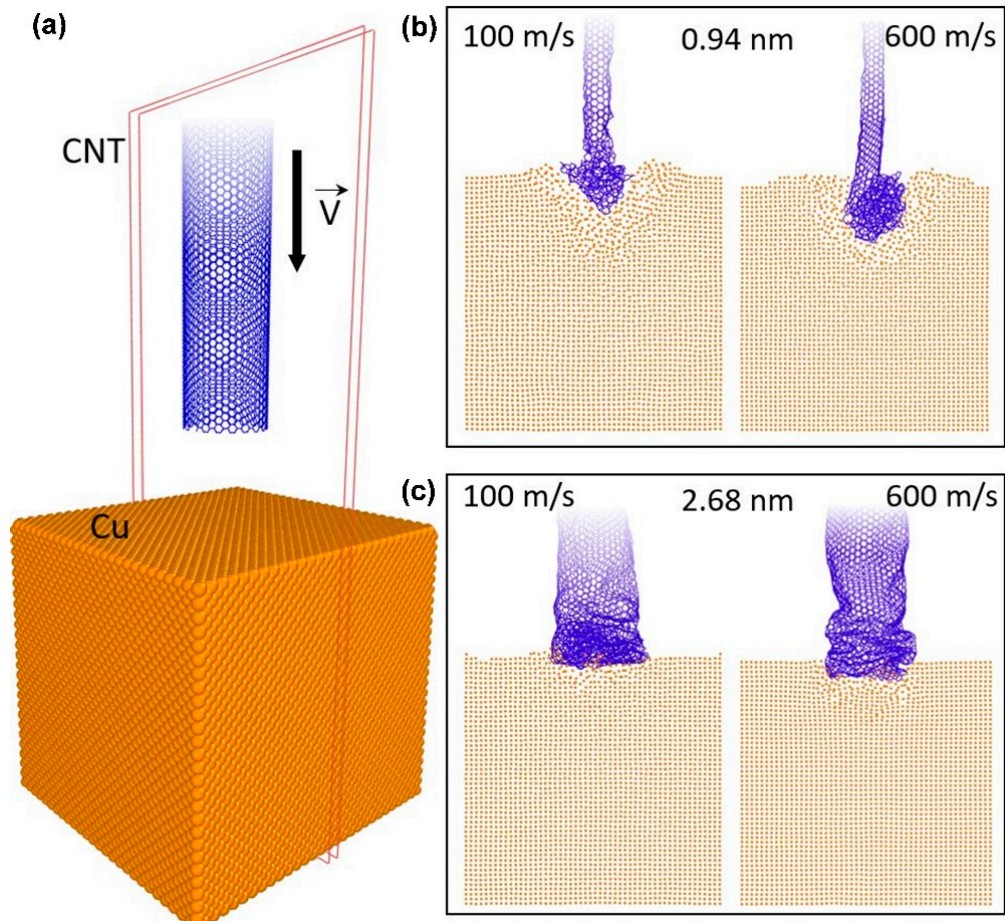

**Figure 4.** Schematic presentation of the molecular dynamic simulation. (**a**) Presentation of the CNT deposition into the copper substrate. (**b**) Interaction of CNT with a diameter of 0.94 nm with copper substrate at the acceleration rate of 100 and 600 m/s. (**c**) Interaction of CNT with a diameter of 2.68 nm with copper substrate at the acceleration rate of 100 and 600 m/s.

Figure 4a illustrates a schematic presentation of the molecular dynamic simulation. Figure 4b,c demonstrate the interaction between the CNTs and copper substrate at two different acceleration rates. The observation revealed that CNT with the shorter diameter penetrates deeper into metallic surface. As expected, the penetration is also deeper for the same CNT at higher acceleration rate. Moreover, the penetration depth into the copper substrate is lower compared to the penetration into aluminum [25]. As well as in case of Al, the saturation of the penetration depth was observed starting from 300 m/s. For the copper, the calculations showed the max penetration depth of 2.5 nm while for the aluminum, the penetration reached 5 nm. This behavior may address the shorter lattice parameter of the copper (3.6 Å) than this of the aluminum (4.04 Å) and subsequent higher stiffness.

Finally, 3D-AFM topography of the obtained surface after deposition of CNTs into copper surface is shown in Figure 5.

Figure 5a,b demonstrate the topography of the pure copper surface and thin film on the copper substrate obtained by the LOD method, respectively. The AFM image presents an irregular surface with vertically oriented CNTs. An almost-homogeneous distribution of CNTs over the surface is observed with the average thickness of the CNT thin film of 50–100 nm. However, several locations on the thin film with thickness of ~150 nm are found, that may be attributed to the deposited CNTs with different length.

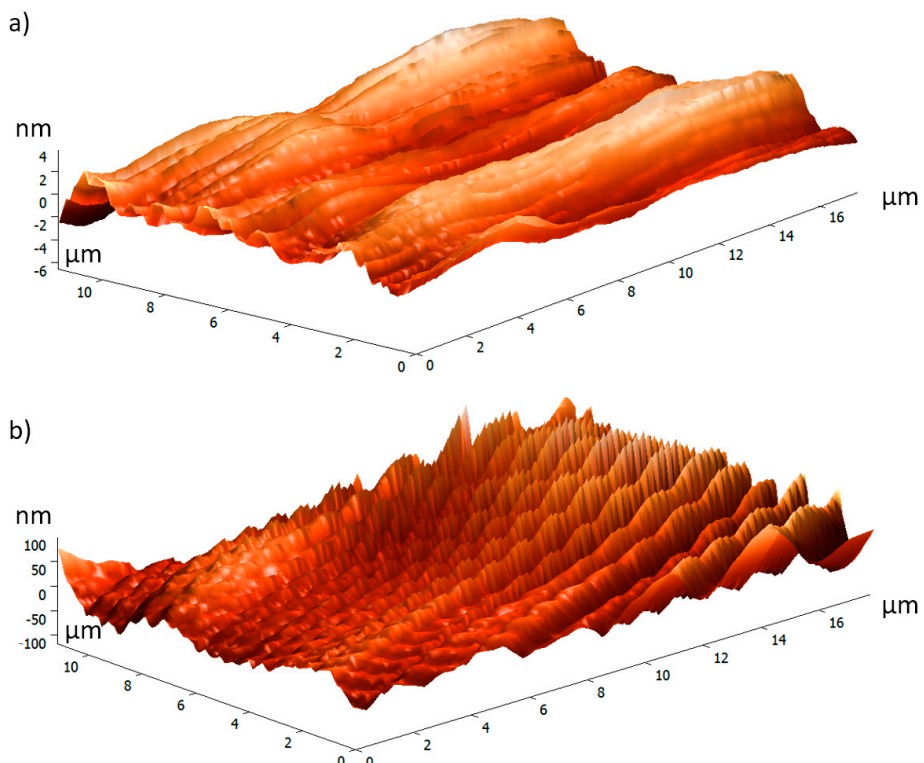

**Figure 5.** AFM-3D images of pure Cu (**a**) and Cu structured with the CNTs (**b**).

It should be mentioned that the visualization of the CNTs at the matrix material surface is very important, and it can support the development of the novel composite structure. The data presented in Figure 6 shows this evidence. A characterization of the possible interaction between the CNTs and copper surface was performed by use of an analysis of the SEM images.

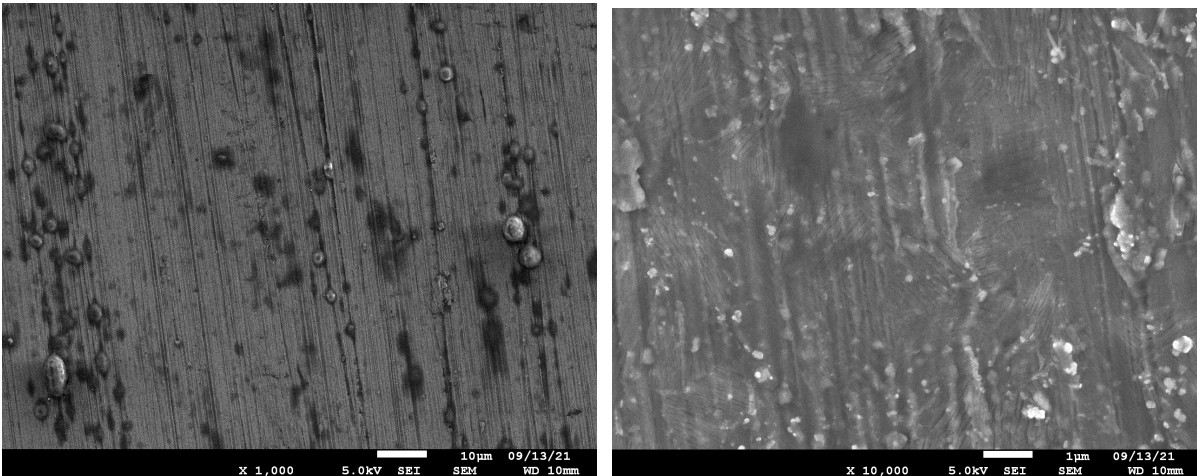

**Figure 6.** SEM images of the pure Cu (**left** figure) and Cu structured with CNTs (**right** figure).

## 4. Conclusions

In this work, the novel perspective composite was shown. The fabrication of CNT thin film on the copper substrate using unique LOD method was presented. Some important properties of this composite were studied and explained. The CNTs-deposited surface exhibited the lower reflectance than the original copper, and microhardness measurements showed the increase in the values of the modified copper by 9%. The wetting phenomenon was also examined. Hence, due to the Lotus effect provided by the CNTs, the CNT thin

film exhibited more hydrophobic behavior, the average CA of 114° and 110° was detected for modified and original samples, respectively. Molecular dynamic simulation revealed that CNT with the lowest diameter and at the highest acceleration rate penetrates deeper into the copper substrate. Moreover, the obtained simulation results showed that the penetration depth was lower in the copper compared to the aluminum substrate, which was addressed to the different crystalline structures of both. The topography examination demonstrated dense thin film with several irregular locations attributed to the different length of the deposited CNTs.

## 5. Patents

To use the LOD treatment of the Cu materials, the following are the details from the patent have been used: Kamanina, N.V.; Vasilyev, P.Ya.; Studeonov, V.I. Optical coating based on oriented in the electric field CNTs for the optical devises, micro- and nanoelectronics under the conditions when the interface: solid substrate-coating can be eliminated, RU Patent 2 405 177 C2 with the priority on 23 December 2008; registered on the State Inventory on 27 November 2010.

**Author Contributions:** Conceptualization, N.K.; formal analysis N.K.; writing—original draft preparation, N.K.; investigation, visualization, A.T.; methodology, resources, simulation, D.K. All authors have read and agreed to the published version of the manuscript.

**Funding:** This research partially supported by the Foundation for the Promotion of Innovation, project No [72598/C1-112174], 2021–2022.

**Data Availability Statement:** The following (LOD block scheme) are available online at https://www.scilit.net/article/6d1a040dd26a2081fe82eb58565844fc, accessed on 17 May 2022.

**Acknowledgments:** N.K. and A.T. would like to thank their colleagues from Vavilov State Optical Institute, LETI University and Nuclear Physics Institute for the useful discussions. Partially this research has been supported by the Innovation Promotion Fund, grant no. [72598/C1-112174] as well as by the SPbETU "LETI" supporting scientific research program via project No.НР/ДЦФиФ-1. D.K. acknowledges the Ministry of Science and Higher Education of the Russian Federation (project No. 01201253304) in the part of molecular dynamics simulations. The calculations were performed using the resources provided by the Joint Supercomputer Center of the Russian Academy of Sciences. Authors would like to acknowledge to Alexey Nashchekin (Ioffe Physical-Technical Institute, RAS, Saint-Petersburg) for the help with the SEM images obtaining.

**Conflicts of Interest:** The authors declare no conflict of interest.

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
