# Peer review of "Nanostructuration Impact on the Basic Properties of the Materials: Novel Composite Carbon Nanotubes on a Copper Surface"

_jcs, doi:10.3390/jcs6060181_

Round 1
Reviewer 1 Report
The presented work is noveltly.
Author Response
Dear Referee!
Thanks a lot for your kind job to consider our paper and to make this positive response!
We will continue to use this laser technique to improve the features of the different materials. It is really innovative method!
English is improved a little bit by me.
Best Regards and be healthy!
Natalia Kamanina
=======================================
Natalia V. Kamanina (Prof., Dr.Sci., PhD
Head of the lab for Photophysics of media with nanoobjects
Vavilov State Optical Institute
Kadetskaya Liniya V.O., dom.5, korpus 2,
St.- Petersburg, 199053, Russia
Professor of the St.-Petersburg Electrotechnical University (“LETI”),
Part-time Leading Researcher at Nuclear Physics Institute (Gatchina)
Job phone: +7 (812) 327-00-95
Fax: +7 (812) 331-75-58 (for N.V.Kamanina)
e-mail: nvkamanina@mail.ru
http://www.photophysics-lab.org/
https://publons.com/researcher/1696479/natalia-kamanina/
https://sciprofiles.com/news-feed
http://rusnor.org/network/webinars/10203.htm
http://www.npkgoi.ru/?module=articles&c=profil&b=7
http://www.nanometer.ru/2007/08/09/liquid_crystal_3905.html
http://www.eltech.ru/ru/fakultety/fakultet-elektroniki/sostav-fakulteta/kafedra-kvantovoy-elektroniki-i-optiko-elektronnyh-priborov/sostav-kafedry
=======================================

Reviewer 2 Report
This manuscript fabricated the composite based on CNTs on the Cu surface by using LOD. Authors also conducted molecular dynamic simulation to investigate the effect of penetration depth and nanotube diameter on the growth of the acceleration rate. The obtained results sound good. However, there are some points, which should be clarity.
1. The English should be carefully revised throughout a manuscript
2. Page 2, line 52, other applications such as solar to electricity (Chem Comm, 2013, 49, 8910; Carbon 116 (2017) 294) or solar to steam (Global Challenges 2018, 2, 1700094) should be referred
3. How to define the composite in this topic?
4. How many layers are in the walls of the used CNT?
5. The morphology of the CNT on the Cu surface should be provided
This manuscript can be considered for publication only when the above-mention questions were especially stressed in the revised manuscript. The referee would like to review a revised version of this paper in the future.
Author Response
Dear Referee!
Thanks a lot for your kind job to consider our paper and to make the useful comments!
I have included my answers in the letter to you and in the text body. All paragraphs included are collared by yellow.
English is improved a little bit. References recommended by you are included in the text body. I have included two of them: 1). Van-Duong Dao and Ho-Suk Choi. “Dry plasma synthesis of a MWNT–Pt nanohybrid as an efficient and low-cost counter electrode material for dye-sensitized solar cells”, Chem. Comm., 2013, 49, 8910-8912. DOI: 10.1039/c3cc42151a. 2). Van-Duong Dao, Liudmila L. Larina, Quoc Chinh Tran, Van-Tien Bui, Van-Toan Nguyen, Thanh-Dong Pham, Ibrahim M.A. Mohamed, Nasser A.M. Barakat, Bui The Huy, Ho-Suk Choi. Evaluation of Pt-based alloy/graphene nanohybrid electrocatalysts for triiodide reduction in photovoltaics. Carbon. 2017, 116, 294-302. http://dx.doi.org/10.1016/j.carbon.2017.02.004. The morphology of the CNT on the Cu surface is provided by SEW images. Figure 6 is added. About your question: How many layers are in the walls of the used CNT? We have used the single wall carbon nanotubes for this LOD method.
Thank you once again!
Best Regards and be healthy!
Natalia Kamanina
=======================================
Natalia V. Kamanina (Prof., Dr.Sci., PhD)
Head of the lab for Photophysics of media with nanoobjects
Vavilov State Optical Institute
Kadetskaya Liniya V.O., dom.5, korpus 2,
St.- Petersburg, 199053, Russia
Professor of the St.-Petersburg Electrotechnical University (“LETI”),
Part-time Leading Researcher at Nuclear Physics Institute (Gatchina)
Job phone: +7 (812) 327-00-95
Fax: +7 (812) 331-75-58 (for N.V.Kamanina)
e-mail: nvkamanina@mail.ru
http://www.photophysics-lab.org/
https://publons.com/researcher/1696479/natalia-kamanina/
https://sciprofiles.com/news-feed
http://rusnor.org/network/webinars/10203.htm
http://www.npkgoi.ru/?module=articles&c=profil&b=7
http://www.nanometer.ru/2007/08/09/liquid_crystal_3905.html
http://www.eltech.ru/ru/fakultety/fakultet-elektroniki/sostav-fakulteta/kafedra-kvantovoy-elektroniki-i-optiko-elektronnyh-priborov/sostav-kafedry
=======================================

Reviewer 3 Report
Dear Authors,
thanks for your paper. It is interesting technique . I found you have already published other two paper on this technique. You showed the powerful of this technique on different substrate aluminium, zinc selenide and zinc sulfide. I reported the references at the bottom.
It is interesting the studios on the surface impact speed . Can the authors provide and show any pictures on Carbon nanotubes (CNTs), before and after the impact on surface? AFM is spectacular technique but here it doesn't carry out so much details on what happened to CNTs. Is it difficult to see their vertical position, clearly.Do you have any SEM images of your samples?
Did you find any fragment of CNTs on copper surface ?Can you exstimate percentage of broken carbon nanotubes on the copper surface? Do they have any effects on properties of surface?
Are you able to cover a selected area of the surface?
Did you measurements the quality of the contact between CNTs and copper? This one is a crucial aspect for many application.
Is it possible to use this technique to design interconnections for high speed technologies or thermal management system?
I push the authors to improve their interesting article: Addition measurements and explanations can improve the quality of paper and mark a difference between this paper and the previous ones.
It is important to show the quality of your composite.
Coatings 2021, 11(6), 674; https://doi.org/10.3390/coatings11060674 Received: 18 May 2021 / Revised: 31 May 2021 / Accepted: 31 May 2021 / Published: 2 June 2021C 2021, 7(4), 84; https://doi.org/10.3390/c7040084
Received: 16 November 2021 / Revised: 3 December 2021 / Accepted: 4 December 2021 / Published: 7 December 2021Author Response
Dear Referee!
Thanks a lot for your kind job to consider our paper and to make the useful comments! I have included my answers to your questions in the letter to you (shown below) and in the text body. All paragraphs included are collared by yellow.
Best Regards,
Natalia Kamanina
=======================================
Natalia V. Kamanina (Prof., Dr.Sci., PhD)
Head of the lab for Photophysics of media with nanoobjects
Vavilov State Optical Institute
Kadetskaya Liniya V.O., dom.5, korpus 2,
St.- Petersburg, 199053, Russia
Professor of the St.-Petersburg Electrotechnical University (“LETI”),
Part-time Leading Researcher at Nuclear Physics Institute (Gatchina)
Job phone: +7 (812) 327-00-95
Fax: +7 (812) 331-75-58 (for N.V.Kamanina)
e-mail: nvkamanina@mail.ru
http://www.photophysics-lab.org/
https://publons.com/researcher/1696479/natalia-kamanina/
https://sciprofiles.com/news-feed
http://rusnor.org/network/webinars/10203.htm
http://www.npkgoi.ru/?module=articles&c=profil&b=7
http://www.nanometer.ru/2007/08/09/liquid_crystal_3905.html
http://www.eltech.ru/ru/fakultety/fakultet-elektroniki/sostav-fakulteta/kafedra-kvantovoy-elektroniki-i-optiko-elektronnyh-priborov/sostav-kafedry
=======================================
Dear Authors,
thanks for your paper. It is interesting technique . I found you have already published other two paper on this technique. You showed the powerful of this technique on different substrate aluminium, zinc selenide and zinc sulfide. I reported the references at the bottom.
It is interesting the studios on the surface impact speed . Can the authors provide and show any pictures on Carbon nanotubes (CNTs), before and after the impact on surface? AFM is spectacular technique but here it doesn't carry out so much details on what happened to CNTs. Is it difficult to see their vertical position, clearly. Do you have any SEM images of your samples?
Did you find any fragment of CNTs on copper surface ?Can you exstimate percentage of broken carbon nanotubes on the copper surface? Do they have any effects on properties of surface?
Thanks a lot for these useful comments. So many recommendations, which we will use in future! Now I have added in the text body the SEM-images (please see Figure 6), which can support the novel composite structures based on CNTs-Cu alloys.
Are you able to cover a selected area of the surface?
Yes we will able to cover the selected area of the surfaces of the materials. For this aim we should develop some masks to select the area. But, our construction in the vacuum chamber connected with the 6 holders for the sample with the diameter of 35 mm. The better for the process to cover all this surfaces (35 mm in diameter).
Did you measurements the quality of the contact between CNTs and copper? This one is a crucial aspect for many application.
Is it possible to use this technique to design interconnections for high speed technologies or thermal management system?
Good suggestion!!! Yes, our LOD technique can be used for this aim as well. But, the engineers should take into account that the IR laser is an expensive tool for production, thus it can be the reason for careful consideration.
I push the authors to improve their interesting article: Addition measurements and explanations can improve the quality of paper and mark a difference between this paper and the previous ones.
It is important to show the quality of your composite.
Coatings 2021, 11(6), 674; https://doi.org/10.3390/coatings11060674 Received: 18 May 2021 / Revised: 31 May 2021 / Accepted: 31 May 2021 / Published: 2 June 2021
C 2021, 7(4), 84; https://doi.org/10.3390/c7040084
Thank you! I have added the paragraph in the paper text body: KBr [23], LiF [24], Al [25] studied and their physical-chemical properties have been improved via application of the laser technique. In each of these cases of material processing, the ratio between the parameters of the elementary lattice of the matrix and the diameter of the nanotubes was taken into account; the expansion rate of the carbon nanotubes was also taken into account when an electric field was applied in the range of 100-600 V/cm.

Round 2
Reviewer 2 Report
I have no more comments on this manuscript
Reviewer 3 Report
Dear Authors thanks for your reply and for the SEM images. Whole improved the quality of the article.
It is difficult to identify carbon nanotubes on the top of the copper surface.
Figure 6, right side, shows something on the surface: are they several bundles of CNTs? it is not clear. The vertical alignment of CNTs is not visible. It is possible to identify something on the top of surface. I suggest to the authors to investigate and show how the CNTs are present on the top of the surface: are they vertical or horizontal? This one is important point for your research. For future publications it becomes relevant to show the position of CNTs on the surface.